# The Many Faces of Lipids in Genome Stability (and How to Unmask Them)

**DOI:** 10.3390/ijms222312930

**Published:** 2021-11-29

**Authors:** María Moriel-Carretero

**Affiliations:** Centre de Recherche en Biologie Cellulaire de Montpellier (CRBM), Université de Montpellier, Centre National de la Recherche Scientifique, CEDEX 5, 34293 Montpellier, France; maria.moriel@crbm.cnrs.fr

**Keywords:** nuclear homeostasis, lipids, genome stability

## Abstract

Deep efforts have been devoted to studying the fundamental mechanisms ruling genome integrity preservation. A strong focus relies on our comprehension of nucleic acid and protein interactions. Comparatively, our exploration of whether lipids contribute to genome homeostasis and, if they do, how, is severely underdeveloped. This disequilibrium may be understood in historical terms, but also relates to the difficulty of applying classical lipid-related techniques to a territory such as a nucleus. The limited research in this domain translates into scarce and rarely gathered information, which with time further discourages new initiatives. In this review, the ways lipids have been demonstrated to, or very likely do, impact nuclear transactions, in general, and genome homeostasis, in particular, are explored. Moreover, a succinct yet exhaustive battery of available techniques is proposed to tackle the study of this topic while keeping in mind the feasibility and habits of “nucleus-centered” researchers.

## 1. Introduction

Along with nucleic acids, proteins, and carbohydrates, lipids are one of the four major biomolecules in living organisms [1], a nomenclature that includes thousands of different species of variable structure [2,3]. Cellular processes are mediated by the concerted action of biomolecules, thus forming complex interaction networks. Yet, while two-thirds of the cell proteome is in constant interaction with multiple lipid species, few efforts have established cellular maps other than protein-protein and protein-nucleic acid interactions [4]. This is in part because profiling lipid-protein interactions fall beyond the working potential of most standard bioanalytical methods, which in turn relates to the partial hydrophobic nature of these molecules.

From a classical point of view, lipids are first recognized as the major components of cellular membranes, which in mammals mostly include the glycerophospholipids phosphatidyl-choline, -ethanolamine, -serine, and -inositol, the sphingolipid sphingomyelin, glycosphingolipids, and cholesterol. Secondly, lipids are also high-energy storage units, whose ATP-producing potential is released during ß-oxidation. Thirdly, lipids are crucial for signaling events, self-organizing in a dynamic fashion to conform rafts. These nanoplatforms can congregate proteins to trigger signals, as classically illustrated during the formation of the immunological synapse in T-cell activation [5]. Lipids also become second messengers after being locally hydrolyzed in response to specific cues, giving rise to diacylglycerol, arachidonic acid, ceramide, and sphingosine, all bearing a strong bioactive potential [6] (Figure 1, grey drawings in the cytoplasm). While these three aspects have been highly studied at the plasma membrane and in the cytoplasm, the potential roles of lipids inside the nucleus are comparatively poorly assessed.

Lipids conform to the nuclear membrane, accounting for genome shielding and homeostasis. Additionally, incipient literature starts to address, mostly in a descriptive manner, the importance of lipids for genome integrity, preservation during DNA damage, sensing, and repair. For example, the specific accumulation of an early intermediate in the mevalonate pathway, namely mevalonate diphosphate, results in dNTP depletion, subsequent replicative stress, and hyper-activation of the Ataxia Telangiectasia and Rad3-related (ATR)-dependent DNA Damage Response (DDR) branch [7]. The synthesis of fatty acids and sterols is reported to be under the control of the Ataxia Telangiectasia-mutated (ATM) axis of the DDR in glioblastoma cells [8] and, reciprocally, saturated fatty acids and cholesterol are described to finetune the DDR activation [9,10]. Yet, the mechanisms underlying these observations remain to be elucidated.

In the following section, ways through which lipids may theoretically impact or have been demonstrated to actually affect nuclear homeostasis are described (a simplified compilation of these roles can be found in blue font in the scheme presented in Figure 1). When justified, how this may uncover a new research avenue in genome stability is speculated. At the end of this section, Table 1 recapitulates the main contribution of different lipid species to different aspects of nuclear physiology in each reported organism. Last, a non-expert-friendly overview of the methodological approaches available to study those aspects is presented, taking care that they remain compatible with the specific needs of genome-related studies.

## 2. How Lipids Impact Nuclear Homeostasis

### 2.1. Directly, via Membrane Composition and Properties

The first and most intuitive contribution of lipids to nuclear homeostasis comes from their structural role as the main constituents of the nuclear membrane. The different membranes of the cell establish their identity through their differential protein and lipid species composition, which in turn derive features such as their curvature and their electrostatic profiles. Lipids whose shape impart a marked curvature to the membrane create a platform capable of attracting proteins. The reactions hosted this way are different from the ones occurring at membranes with a very negative charge yet low curvature [11]. In this sense, the nuclear membrane, for example of liver cells, contains a very rich profile of polyunsaturated fatty acids [12]. This provides a loose packing environment that is permissive for the insertion of a specific subset of proteins, which can be exploited to establish the very characteristic inner nuclear membrane (INM) proteome [13]. The lipid composition of the nuclear membrane also conditions the assembly and configuration of nuclear pore complexes [14]. A particularly important contribution goes for the sphingolipid hydrolase SMPD4 [15], which, by releasing ceramides specifically in proximity to nuclear pores, could either promote the local concave membrane curvature needed to insert nuclear pore complexes [16] or act as enzymatic cofactors (see next section). Further, the production of very-long-chain fatty acids suppresses ruptures in the fission yeast nuclear membrane [17] and is key in sustaining the extreme curvature of the membrane at the sites of nuclear pore complexes insertion [18]. Lipids alleviate the stress derived from defects during nuclear pore complex assembly [19], and the overexpression of Ole1, which increases the presence of unsaturated acyl chains, solves defects in nuclear envelope sealing in *Schizosaccharomyces japonicus* [20].

The nuclear membrane lipid profile also supports its role in genome shielding. For example, the accumulation of long-chain sphingoid bases suppresses the aberrant nuclear membrane defects induced by aneuploidy, both in budding yeast and human cells [21]. In this sense, we recently reported that lipid stress is elicited at the INM by the widely used genotoxic agent Methyl methanesulfonate [22]. Its alleviation occurs through membrane deformation or by the emission of membrane-derived structures into the nucleoplasm [22], two processes that can affect ploidy control [21] or DNA repair [23], respectively. The excess of polyunsaturation in the chains of phosphatidylcholine can also be sensed by the apical kinase of the DDR ATR in a DNA damage-independent manner. Activated by this stimulus, ATR phosphorylates p53, which in turn halts the cell cycle and can eventually lead to apoptosis [24]. This sensing by ATR of lipid membrane properties may be at the basis of its recently reported ability in mechanosensing [25,26]. Last, restricting phospholipid availability by the use of the fatty acid synthase inhibitor cerulenin provokes an extremely round nuclear morphology [27]. As a consequence, it stems that the plasticity and other biophysical properties of the nuclear membrane, as ruled by lipids, will virtually impact every nuclear transaction requesting the membrane or the nuclear pores. For example, the anionic glycerophospholipid phosphatidylserine contacts chromatin to promote the re-nucleation of the nuclear envelope after mitosis [28]. Other nuclear events impacted by lipids include the coordination of transcription and replication, in which DNA needs to be dynamically anchored to and dislodged from the nuclear membrane [29]. Telomeres in *Saccharomyces cerevisiae* are also anchored to the nuclear membrane, and defects in the lipid-controlled ESCRT (endosomal-sorting complexes required for transport) machinery result in telomere shortening [30]. In the same line, the nuclear membrane influences the establishment of chromosome territories of the epigenome, and its maintenance [31,32,33], the DDR [9,34], and DNA repair [23,35,36]. Therefore, nuclear membrane shaping by lipids conditions nuclear events.

### 2.2. Directly, through Protein Lipidation

Lipidation is a powerful post-translational covalent modification with the ability to alter the sub-cellular localization of proteins, modulating their catalytic activity, provoking their aggregation, and changing their half-life as well as their interacting partners. For example, palmitoylation can compete with ubiquitination at specific lysines, thus protecting the modified protein from degradation [37]. Lipidation can be actively mediated by enzymatic conjugation or occur in a passive way. Its modalities vary depending on the lipid moiety that is conjugated (a fatty acid, a cholesterol molecule, or an isoprenoid) and on the type of amino acid to which it becomes esterified, namely lysines, serines, cysteines, or glycines [38]. The most frequently conjugated fatty acids include the irreversible addition of myristate at the N-terminus of proteins and the reversible esterification of palmitate at inner lysines and serines of the targets. We are familiar with nuclear proteins being controlled by different post-translational modifications, such as phosphorylation, ubiquitination, or sumoylation. Yet, we are less aware of the potential control that lipidation may exert, and the concerned nuclear proteome. Notably, nuclear proteins, as the replicative helicase MCM and the DNA replication factor C, have been found to be palmitoylated in HeLa cells, although the physiological role of this modification remains unassessed [39]. Several variants of the histone H3 are palmitoylated at cysteine 110 [40], and palmitoylation of histone H4 at serine 47 is important to regulate transcription [41]. Further, the palmitoylation of Rif1 permits its anchor to the nuclear membrane, which warrants its role in heterochromatin formation [42] and in mediating accurate DNA double-strand break repair [43]. 

A relevant example of the importance of nuclear proteins’ lipidation concerns the farnesylation of Lamin A. The *LMNA* gene encodes four lamins (A, C, CD10, and C2) via alternative splicing, of which lamin A and C are expressed ubiquitously. Lamin A precursor, prelamin A, is farnesylated, a lipidation permitting its association with the nuclear membrane. Cleavage by the endoprotease ZMPSTE24 gives rise to a mature form that lacks the lipidated moiety and is released from the membrane. A pathological de novo single-base substitution (c.1824C > T) occurring within the *LMNA* exon 11 gives rise to an aberrant protein, termed progerin, that can be lipidated yet loses the site for the proteolytic cleavage, thus remaining permanently modified [44]. This mutation acts in a dominant fashion, is at the base of misshaped nuclei, heterochromatin loss, delayed DDR, mislocalization of nuclear proteins, and at the organism level, gives rise to Hutchinson Gilford Progeria Syndrome [45,46].

Interestingly, passive lipidation of proteins may also be key in the nucleus. In fact, thousands of these acyl modifications found in vivo occur at a very low stoichiometry, thus making enzyme-driven, regulated lipidation a rather unlikely event. In contrast, slow, nonenzymatic reactions are more plausible. Indeed, acyl-CoA thioesters can react with nucleophilic cysteine and lysine residues. The exposure of such residues in the surface of proteins suffices to undergo this spontaneous modification. Given that cysteines are rarely exposed to protein surfaces, nonenzymatic acylation mostly impacts lysines [47]. If not antagonized by active de-acylation, nonenzymatic acylation becomes a *bonafide* “carbon stress”, in which its detrimental impact is suggested by its negative selection throughout evolution at hundreds of sites prone to suffer spontaneous lysine acylation [48]. This way, active de-acylation prevents the deleterious inactivation of the proteome due to undesired spontaneous acylation while restoring the pool of available acyl-CoAs [49]. The de-acylation activity, in general, and in front of nonenzymatic acylation, in particular, relies on sirtuins [50]. In the context of the nucleus, the danger of carbon stress is illustrated by the essentiality of SIRT6 for nuclear homeostasis. SIRT6 is a predominantly nuclear protein [51], functioning as an ADP-ribosylase and as a NAD^+^-dependent de-acylase, both for short acyl groups (for example, acetyl) and for long-chain fatty-acyl ones [52]. The absence of SIRT6 alters telomere position effect [53] and DNA damage sensing at an early step preceding ATM signaling [54]. It also affects DNA double-strand break repair by destabilizing DNA-PK [55] and hampers the recruitment of the chromatin remodeler SNF2H to DNA break sites [56]. Further, SIRT6 appears to be instructed by Lamin A in deacetylating histones [57]. Yet, in spite of being implicated in these and many other cellular functions, the precise manner in which its activity in DNA damage repair is exerted remains largely elusive [52]. Since, upon detection of long-chain fatty acids, SIRT6 catalysis is stimulated [58], it is both tempting and parsimonious, in order to explain all its aforementioned roles, to invoke its role as a “nuclear de-lipidation” cleaner whose activity keeps a healthy nuclear proteome.

Last, passive lipidation of proteins may occur with lipid peroxidation products (oxoLPP), giving rise to modified proteins of compromised function. Importantly, nuclear proteins related to nucleopore function, transcription, and splicing, as well as DNA damage sensing and repair, have been reported as specific targets of such a modification upon high-fat cell loading [59].

### 2.3. Directly, through Non-Covalent Lipid Binding 

Long-chain fatty acids are poorly soluble in aqueous environments and, to increase their solubility and facilitate their movement through the cytoplasm, utilize the Fatty Acid-Binding Proteins (FABPs) [60], abundant members of an ancient and conserved multigene family [61]. In parallel, long-chain fatty acids regulate the expression of several genes involved in lipid metabolism thanks to ligand-dependent transcription factors, such as the nuclear peroxisome proliferator-activated receptor, PPAR [62,63,64,65]. It was therefore suggested that FABPs impact gene expression by transporting bioactive long fatty acids to the nucleus and delivering them to the transcription factors. FABPs such as CRBP (cellular retinol-binding protein), cellular thyroxine-binding protein (CTBP), and liver FA binding protein (L-FABP) have been evidenced through in vitro approaches using isolated nuclei to implement such a transport ([66,67,68,69], to cite a few). Recently, the in vivo localization of Fabp1b.1 and Fabp2 was reported within the nuclei of enterocytes in zebrafish, in particular as excluded from condensed chromatin [70]. Thus, in nuclei, FABPs exert a transport and delivery function whose best-recognized goal is to implement ligand-induced transcription. Yet, this does not rule out that fatty acids (FA) being transported to and moving within the nucleus by FABPs may have additional destinies.

Reciprocally, FABPs ability to bind FA may not be simply restricted to a transport-delivery role. In fact, FABP1 is key in the detoxification of dangerous fatty acids, either because of their excess or because of their harmful activities [71]. This may be important when it comes to electrophilic FA, such as FA nitroalkenes. These endogenously detectable products arise after nitric oxide reacts with unsaturated fatty acids during nitrite-dependent metabolic and inflammatory events. Given their electrophilic nature, FA nitroalkenes mediate post-translational modifications of hyper-reactive nucleophilic cysteine thiols in proteins (as mentioned in the previous section). These types of molecules harm central DNA repair players such as RAD51, for example, by abolishing its ability to bind single-stranded DNA [72]. In agreement with the potential role of FABPs in detoxifying the nucleus, FABP4 binds electrophilic FA at this precise location [73].

Last, an exciting and underexplored scenario is the possibility that other nuclear proteins apart from FABPs bind FA, thus being regulated, redirected, structured, or activated this way. For example, the release of stored fatty acids (i.e., lipolysis of triacylglycerols) triggers a transcriptional response, stimulated by sirtuin 1 (SIRT1), aimed at coupling FA release with their ß-oxidation [74]. It was recently found that perilipin 5 (PLIN5), a protein residing on cytoplasmic lipid droplets, the main FA cellular store (see next section), can bind mono-unsaturated FA released during lipolysis and traffic them to the nucleus, which stimulates SIRT1 catalysis, therefore enhancing the aforementioned transcriptional program [75]. Another example concerns the tumor suppressor and master of nuclear integrity BRCA1 (BReast CAncer 1), which harbors, at aminoacid residues 1664–1696 within the first BRCT repeat, a lipid-binding motif. In the 3D structure, these residues are solvent-accessible and share strong homology to FABP pockets. This ability has been proposed to mediate the relative distribution of BRCA1 between the nucleus and the cytoplasm [76]. To finish, we recently predicted in silico the presence of one FA-binding pocket in Nup157 and in Nic96, two nucleoporins necessary to anchor and scaffold the nuclear pore complex. Both pockets are located at strategic sites mastering the 3D conformation of these nucleoporins. Genetic experiments and fluorescence microscopy suggested that FA binding may help solubilize these proteins to escort them for assembly at the nuclear membrane [77]. Thus, FA binding bears a strong potential for ruling very divergent nuclear phenomena.

On a different note, ceramides arise as the major metabolite after sphingomyelin hydrolysis at the plasma membrane in response to chemical and physical stresses, as well as agonist binding. Ceramides generated this way diffuse from this site of cleavage to act as second messengers, tuning trafficking, inflammation, proliferation, differentiation, and apoptosis [78]. This regulatory activity is exerted through their non-covalent binding to target proteins, both in a stimulatory and an inhibitory manner. Among such proteins, Protein Phosphatases 1 and 2A (PP1 and PP2A) were identified as ceramide-activated using short length, cell-permeable C_2_-ceramide [79,80,81]. Later on, their ability to stimulate these phosphatases was proven using more physiologically relevant long-chain ceramides, such as D-erytro-C_18_-ceramide [82]. This stimulatory role of ceramides on phosphatases may be key upon genotoxic stress, as various phosphatases have been implicated in silencing the DDR. This includes PP2C (Ptc2/Ptc3) and PP4 (Pph3-Psy2) in the recovery from a double-strand break [83,84], as well as PP1 (Glc7) and again PP4 (Pph3-Psy2) to permit recovery after stress during replication [85,86,87,88]. PP2A also shows activity in dephosphorylating important genome integrity surveillance players such as ATM, p53, Chk1, Chk2, and Rad53 [89,90,91].

Importantly, to tolerate the replication stressor hydroxyurea, *S. cerevisiae* cells are dependent on the genes coding for Isc1 (the yeast homolog of mammalian neutral sphingomyelinases, which catalyzes the hydrolysis of complex sphingolipids, thus releasing dihydro- and phytoceramides), Sur4 (Fatty Acid elongase), Lag1 and Lac1 (ceramide synthase), and cannot tolerate the overexpression of the phytoceramidase Ydc1 [92]. Overall, this speaks of an acute dependency on ceramides to tolerate hydroxyurea. Later work demonstrated that, specifically, C_18:1_-phytoceramide activates Cdc55, a regulatory subunit of PP2A, which culminates with cell cycle resumption [92]. A further contribution of ceramides to the tolerance to hydroxyurea comes from stimulated PP2A de-phosphorylating of the DDR effector Rad53 [89]. Thus, ceramides are necessary to de-activate the checkpoint elicited by replication stress. Of note, there exist many species of ceramides, distinguished by the length and the unsaturation degree of their fatty acyl group and the hydroxylation of the sphingoid base. Recent work has expanded the catalog of ceramides implicated in response to hydroxyurea stress [93]. As such, ceramides could modulate enzymatic activities other than phosphatases, processes other than the DDR, and in response to nuclear stresses other than hydroxyurea.

### 2.4. Directly, as Nucleating and Scaffolding Platforms inside the Nucleus

Phosphoinositides (PIP) are a subset of phospholipids with inositol as the polar head that can become phosphorylated in all possible combinations at positions 3, 4, and 5, giving rise to seven possible variants. These molecules are present in the nucleus, and their levels are responsive to multiple stimuli, therefore suggesting their potential role in regulating specific nuclear functions [94]. Both the synthesis of PIP at the INM and their intranuclear presence are documented [95,96]. It is even possible that their synthesis occurs within the nucleoplasm provided that, as soon as they emerge from their synthesizing enzyme, they immediately engage in interactions with surrounding proteins and nucleic acids [97]. This is due to their poor solubility, which does not render it intuitive to imagine how they manage to be stabilized or to diffuse in the nucleoplasm. One option is that their FA tails conform micelles or establish interactions with nuclear proteins that effectively mask their hydrophobic features [94]. For example, their apolar tails can be bound by the steroidogenic factor 1 (SF-1) while the inositol head group remains exposed to establish interactions [98,99]. Excitingly, SF-1 has been implicated in the process through which PIP affects DNA damage sensing [98].

The poor definition of PIP nuclear targets has hindered the characterization of the nuclear pathways they control. Targeted and global proteomic studies have established lists of potential PIP-interacting proteins, pointing at transcription, chromatin remodeling, and mRNA maturation [100]. For example, ING2, one member of the conserved inhibition of growth (ING) family, associated with the modulation of histone acetyl transferase and deacetylase complexes [101], was described as a *bonafide* nuclear receptor for PI(5)P. This interaction regulates the ability of ING2 to activate p53 and p53-dependent apoptotic pathways [102]. Since then, chromatin regulation has been well-established to be under PIP influence [103]. Mechanistically, PIP provides negative charges locally at the regions where they are embedded, thus creating domains for interaction with positively charged proteins. Of note, protein acetylation can neutralize the charge associated with its target lysine, thus erasing this positive charge [104]. It was very recently shown that, within proteins interacting with membranes, acetylation occurs more frequently at the specific protein surface responsible for the interaction with negative membrane patches, thus abolishing its interaction with these patches in a reversible manner [105]. Given that acetylation is a common post-translational modification regulating nuclear proteins, in general, and histones in particular, it will be worth exploring how PIP may play on the equilibrium of chromatin attachment to/detachment from the nuclear periphery.

An additional yet less recognized picture emerges when exploring the contribution of PIP as scaffolding platforms. In particular, the involvement of PI(4,5)P_2_ in the organization of membrane-less nuclear compartments. This definition concerns sub-nuclear domains with a functional identity self-structured through the nucleation of their component molecules without being contained by a shielding membrane. These include splicing speckles, nucleoli, and DNA repair foci. In all cases, PI(4,5)P_2_ underlies their organization [98,106], and its absence has negative functional consequences in splicing implementation [107], RNA polymerase I transcription [108], and sensing of DNA damage [98], respectively. In the same line, a very recent work reports the presence of PI(4)P in close contact with the nuclear lamina, the nucleoli, and the nuclear speckles, as well as scattered through the nucleoplasm. Identification of its associated proteome points at replication, transcription, as well as mRNA and rRNA processing [109].

### 2.5. Directly, as Chaperones

A yet different contribution of lipids to nuclear homeostasis concerns their ability to act as chemical chaperones. Lipid species of the sterol family have demonstrated a palette of dissolving activities towards protein aggregates. For example, an endogenous sterol refractory to identification was found responsible for the reversible disassembly of misfolded proteins stored within inclusion bodies, and authors could mimic the same disaggregation power by externally providing 25-hydroxycholesterol [110]. In the same line, 25-hydroxycholesterol and its related molecule lanosterol promote the chemical disassembly of aggregated proteins in vitro and in vivo [111,112,113]. The nucleolus harbors a complex and dense meshwork of proteins and RNA and a well-demonstrated link with protein aggregation, both of pathological nature [114] or with protective purposes [115]. In both cases, reversibility of nucleolar aggregates is a *sine qua non* condition to regain homeostasis. It would be very exciting to explore the potential of these molecules in this specific context. On another chaperoning set-up, cholesterol was recently shown in vitro to promote chromatin compaction through the dehydration of nucleosomes [116]. Further, cholesterol complexed with single-stranded DNA can instruct DNA bending in order to hide the exposed end [117], opening a myriad of regulatory roles, in particular during DNA repair. Excitingly, lipid membranes can also act as an RNA organization platform in an RNA sequence-, length- and structure-dependent manner [118]. This introduces a novel layer for ribo-regulation, but also opens unexplored avenues in how lipids, near both the nuclear membrane or the nucleoplasm, could impact RNA transcription rates, splicing, packing for export, or selective storage within nuclear speckles.

### 2.6. Indirectly, through the Titration and Supply Activity of Lipid Droplets

Lipid droplets (LD) are cytoplasmic organelles born from the endoplasmic reticulum conformed by a neutral lipid core surrounded by a phospholipid monolayer in which proteins intercalate directly, attach by lipidation moieties or interact through stably anchored partners. They constitute a central store of lipids with either reserve functions to protect them from harm or, reciprocally, to protect the cell from lipid excess (lipotoxicity) [119]. By virtue of the dynamic and variable proteome that can decorate them, LD are now also recognized as a selective protein reservoir that can render such proteins available, or not, at other cellular compartments. The protein composition of LD is exquisitely dynamic and highly responsive to different cues, including the type of lipids they store [120]. In agreement with LD location and main role, the bulk of reported LD-sitting proteins is mostly cytoplasmic. Nevertheless, the published LD proteome datasets are numerous (a comprehensive list is compiled in [121]), and examples of nuclear or nucleus-related factors exist. For example, histones are LD-harboured in different species [122,123,124]. In the *Drosophila* developing embryo, their timely release allows an accurate division pattern [125,126], while in other species, the significance of this storage remains unassessed. Since, conceptually, “division pattern” is intimately related to “replication”, and replication is extremely sensitive to histone levels [127], it is very tempting to speculate that LD may serve as a depot to fine-tune the number of histones supplied to the nucleus at a given time. This may be of special significance during sudden halts in replication, such as during replication stress, when LD could buffer, in a fast manner, a histone excess that otherwise would be deleterious. Recent reports also declare LD as particularly well suited for interacting with transcription factors, such as MLX in human cells [128], MLDSR in bacteria [129], or Opi1 in budding yeast [130]. This is even more stimulating when considering that LD has been reported to bind nucleic acids, in particular RNA [131], thus making emerge the theoretical possibility of a whole transcriptional regulatory network.

To add to this, we recently exploited published LD proteomes from *S. cerevisiae* cells [120] to validate thus enlarge the catalog of nuclear proteins whose presence onto LD is physiologically relevant. While reported hits such as condensin or cohesin complex subunits could not be validated [132], we discovered the ability of cytoplasmic LD to harbor nucleoporins in a manner concerted with cell metabolism [77]. Thus, active cycling is coupled to nucleoporin release from LD to conform active pores and, reciprocally, storage of nucleoporins onto LD permits nuclear pores to modify their stoichiometry under low-cycling conditions [77]. Importantly, we also demonstrated that the inability to use LD as a buffer depot when nucleoporins are in excess overloads the nuclear membrane with nuclear pores and expands it. Further, we revealed the notion that aberrantly formed LD sequester karyopherins and their cargos, thus depleting them from the nucleus [77]. The cytoplasmic accumulation of nucleoporins in bodies positive for the PML protein, named CyPNs, had been proposed to represent a nucleating center for the partial pre-assembly of nuclear pore complexes [133,134,135]. This analogy raises the exciting possibility that at least a subset of LD works as a “montage” factory to create nuclear pore rudiments, which could facilitate their subsequent insertion into the nuclear membrane. This would agree with the recently discovered role of LD in preventing nuclear pore complex assembly stress [19]. Furthermore, mRNAs coding for nucleoporins were recently reported to be targeted to the proximity of the nuclear pore to couple nucleoporin translation with pore assembly [136]. Given the connection of LD with both nuclear pore rudiments [77] and RNA binding [131], it would be interesting to assess whether LD contributes to this nucleoporin mRNA targeting.

A specific mention goes for nuclear LD (nLD), which, in analogy to their cytoplasmic counterparts, are also born from the endoplasmic reticulum, although in this case from the INM subdomain and towards the nucleoplasm [137]. Being less frequent and recently discovered, their function(s) remains a mystery as of today. They are more protein- and sterol-rich than their cytoplasmic counterparts [138], and their birth sites are specified by specific protein and lipid combinations to overcome the negative curvature of the inner side of the nuclear membrane [139,140,141,142]. Of note, we recently reported their birth in response to the genotoxic agent Methyl methanesulfonate [22]. The very first report on the identity of their proteome, just published, also reveals the presence of multiple histones [143]. This could point at a hub controlling the stock of histones to be supplied for replication or again as an assembly platform. Given that LD can interact with RNA [131] and that transcription factors have been reported as assembled onto nLD [130], again, a role in organizing transcription can be invoked. Last, and again in analogy with the role of cytoplasmic LD [110], a non-exclusive function could support the slow release of their lipid contents at given sub-nuclear domains with chemical chaperoning purposes.

### 2.7. Indirectly as a By-Product of Metabolic Transactions

Indirect impact on nuclear transactions is also achieved because of the metabolism of lipids outside the nucleus. For example, acetyl-CoA provides a well-established link between energy metabolism and chromatin regulation [144,145]. In this context, *de novo* FA synthesis uses acetyl-CoA as a substrate, which directly competes with the need of this substrate for histone acetylation. This is why diminishing the rate of FA synthesis through reduced Acetyl-CoA-carboxylase (ACC1) expression increases global histone acetylation and gene expression [146]. Reciprocally, the same effect can be achieved if FA oxidation, which releases acetyl-CoA, is stimulated [147]. From this, it stems that both physiological and pathological alterations in mitochondrial activity have an immediate effect on the epigenome [148]. This concerns tricarboxylic acid intermediates such as not only acetyl-CoA, but also α-ketoglutarate, for they are substrates of histone acetyltransferases and demethylases, respectively. For the same reason, the lack of consumption of S-adenosylmethionine during the defective synthesis of phosphatidylcholine renders histones the preferred sink for this excess of methyl groups, thus fully altering the epigenetic landscape [149].

### 2.8. Indirectly, When DNA-Related Proteins Possess an Additional Role in Lipid Metabolism

That a protein can accomplish two different tasks within the cell is no big surprise. Prp19 was first identified as Pso4, a gene required in *S. cerevisiae* to support error-prone recombinational DNA repair [150]. This role was later extended to human cells [151,152]. Prp19, under this name, was discovered as part of a complex, the NineTeen Complex, that functions during the catalytic activation of the spliceosome [153,154,155]. Likely linked to the defects in splicing caused by Prp19 lack, Prp19 has also been reported as a transcription elongation factor [156]. Mechanistically, the Prp19 complex harbors an E3 ubiquitin-ligase activity that it exerts onto the spliceosome [157]. Its precise function on DNA damage remained elusive for a long time until PRP19 was found to impart the ubiquitylation of the single-stranded DNA coater RPA in response to DNA damage, therefore optimizing the signaling by ATR [158]. Put together, these data pointed at PRP19 as a protein with an RNA-processing role in undamaged cells that switched to a DNA damage sensor upon DNA damage. However, PRP19 was also found in the cytoplasm as part of the proteome of LD isolated from cultured 3T3-L1 adipocytes, its absence leading to a loss of stored triacylglycerols. A pool of relevant proteins, important for the maintenance of LD size and contents, dropped from LD in the absence of PRP19, including perilipin, stearoyl-CoA desaturase-1, acyl-CoA diacylglycerol acyltransferase-1, and glycerol-3-phosphate acyltransferase. These data suggested that PRP19 participates in the maturation of LD and associated fat storage, thus warranting the process of adipocyte differentiation [159]. Amino acids 167–250 ensure PRP19 recruitment to LD, while the 1–166 domain ensures its recruitment to the nucleus [160]. While awaiting a unifying explanation, one could envision that PRP19 may be, also by means of its ubiquitin conjugation activity, important to shape the LD proteome. Indeed, ubiquitination-driven modulation of the LD proteome has been described [161,162]. Importantly, if PRP19 levels in the cell are limiting, it stems that situations leading to the unscheduled growth of LD, such as during obesity, could entail a risk of PRP19 titration from its nuclear duties.

Another example of a nuclear factor with a dual role in the nucleus and the cytoplasm is the already mentioned tumor suppressor BRCA1. BRCA1 works in DNA repair and replication fork stability, but it is also present in the cytoplasm by means of a nuclear export sequence [163]. The natural equilibrium of this distribution is altered in cancer, with a predominant cytoplasmic pattern in 36% of the cases [164]. In particular, cytoplasmic mislocalization of BRCA1 is associated with a subgroup of clinically relevant cancer mutations [165]. One important role of BRCA1 in the cytoplasm concerns the stabilization of the phosphorylated form of ACC1. ACC1 is the rate-limiting enzyme for long-chain FA synthesis, and its phosphorylation on serine 79 by the AMP-activated protein kinase, AMPK, inactivates it [166]. Thus, the stabilization of ACC1 phosphorylation by BRCA1 sustains its inactivity, thus limiting lipid synthesis [167,168]. It is therefore postulated that the BRCA1 mutations that allow ACC1 de-phosphorylation underlie the high FA synthesis that characterizes pathological states, such as in cancer [169]. Nonexclusively, BRCA1 has been reported to bear a lipid-binding pocket capable of driving its interaction with calcium channels at the endoplasmic reticulum. This way, BRCA1 can regulate calcium release from the endoplasmic reticulum to the cytoplasm both in a basal way, and abruptly in response to apoptotic cues [76]. Together, BRCA1 potential to bind and control lipids suggests a provocative regulatory loop by which, in the event of sensing an excess of FA, this may promote its exit from the nucleus, stabilization of the phosphorylated form of ACC1, its inactivation, thus finally, the decrease in the FA pool. Further, an increase in cues invoking BRCA1 in the cytoplasm will deprive the nucleus of its functions in DNA repair and replication fork protection [170], creating a BRCAness phenotype [171].

**Table 1 ijms-22-12930-t001:** Compendium of demonstrated roles of lipid species on nuclear biology.

Action	Lipid Species	Organism	Impact	References
Structural: nuclear membrane composition	polyunsaturated fatty acids	rat liver cells	INM proteome shaping	[12]
	sphingolipids, ceramides	human cells	insertion of nuclear pores	[15,16]
	long-chain fatty acids	*S. japonicus*, *S. cerevisiae*	prevents ruptures	[17,18]
	unsaturated fatty acids	*S. pombe*, *S. cerevisiae*	supports sealing	[20]
	long-chain sphingoid bases	*S. japonicus*	alleviates aneuploidy-related deformation	[21]
	phosphatidylserine	human cells	membrane reformation after mitosis	[28]
	low phospholipid availability	*S. cerevisiae*	extremely round nucleus	[27]
Structural: scaffolds within the nucleus	phosphoinositides (PI(4,5)P)	human cells	scaffolding of membrane-less bodies (splicing speckles, nucleoli, DNA repair foci)	[98,106,107,108,109]
Signaling	sphingolipids, ceramides	human cells	ATR activation	[24]
	mevalonate diphosphate	human cells	ATR hyperactivation	[7]
	saturated fatty acids	human and murine cells	attenuation of the DDR	[9]
	cholesterol	human cells	supports Chk1 activation upon DNA damage	[10]
(de)Lipidation	palmitoylation of histone H4^Ser47^	murine and human cells	transcription regulation	[41]
	palmitoylation of Rif1	*S. cerevisiae*	heterochromatin formation, DNA repair	[42,43]
	farnesylation of Lamin A	human cells	if constant, pleiotropic genome instability	[45,46]
	acyl groups	human cells	pleiotropic genome instability (i.e., in the absence of the SIRT6 deacylase)	[53,54,55,56,57]
Titration	lipid droplets	*S. cerevisiae*	nucleoporin availability	[77]
		human cells, *S. cerevisiae*, *Rhodococcus jostii*	transcription factors availability	[128,129,130]
		*Drosophila*, *S. cerevisiae*, *Plasmodiophora brassicae*, and	histone buffering	[122,123,124,125]
		human mast cells	RNA distribution	[131]
Metabolic by-products	acetyl-CoA α-ketoglutarate	*S. cerevisiae* and human cells	gene expression patterns alteration	[146,147,148,149]
Co-factors	fatty acids	murine, human and zebrafish cells	bind, activate and translocate transcription and DNA repair factors and nucleoporins	[66,67,68,69,70,75,76,77]
	ceramides	*S. cerevisiae*, murine, and human cells	activators of DDR phosphatases (tolerance to genotoxic stress and cell cycle progression)	[79,80,81,82,83,84,85,86,87,88,89,90,91,92,93]
	phosphoinositides (PI(5)P)	human cells	drives ING2 for histone modification	[102]

## 3. How to Tackle Their Study

### 3.1. Studying Lipidation 

While internal palmitoylation sites can be predicted by bioinformatics approaches, software-based prediction of N-terminal myristoylation has not proven of high accuracy or sensitivity (reviewed in [38]). Further, the standard proteomics approaches aimed at detecting post-translational modifications do not work for lipidation because the relatively large and very hydrophobic nature of lipidation modifications restricts ionization of peptides during mass spectrometric analysis, as well as are insensitively labeled by radioactive isotopes [172]. One relatively recent and efficient option to assess lipidated proteins takes advantage of “click” chemistry, which allows the selective coupling of two functional groups (alkyne and azide) in biological samples. In more detail, a commercially available “clickable” lipid (that is, conjugated to an alkyne) can be fed to the cells. The click reaction permits the in vivo coupling of this lipid to a subsequently added reacting azide, itself coupled to biotin. Streptavidin-driven biotin purification allows proteome analyses by Western blotting or classical mass spectrometry (Figure 2, upper left panel) [39]. 

Furthermore, it is possible to use a cleavable azide molecule which, upon digestion, will leave a hydrophilic and charged tag on fatty-acylated peptides. This method increases both the hydrophilicity and ionization of the peptides, optimizing their detection by mass spectrometry, thus enabling the identification of the lipidation sites on the protein [173]. However, despite its versatility and power, one should keep in mind that this procedure operates via metabolic labeling, which interferes with basal metabolism and may alter normal cell processes.

Apart from proteomic studies, the alternative coupling of the reacting azide to a fluorophore permits to image, at the desired time-points, the kinetics and subcellular localization of the lipidation reactions (Figure 2, bottom left panel) and [174,175]. Yet, this technique is limited in that it monitors the whole proteome subjected to that specific lipidation at that given timepoint. A more specific approach combines Proximity Ligation Assay and the click reaction. This way, a specific primary antibody recognizes the protein whose lipidation is under study, while the lipidation moiety is detected by another primary antibody against the biotin tag. Appropriate secondary antibodies coupled to matching oligonucleotides permit rolling circle-based DNA amplification using fluorescent nucleotides, thus providing signals that emanate from the lipidated protein (Figure 2, bottom left panel) [176,177].

### 3.2. Assessing Non-Covalent Lipid Binding to Proteins

#### 3.2.1. Use of Strips, Beads, and Liposomes 

The ability of proteins to bind lipids can be evaluated through the simple presentation of the lipid to the purified protein. The choice, and the challenge, is how the lipid will be presented. A very simple method consists of spotting the lipid species of interest on a hydrophobic membrane (lipid strips), onto which a solution containing the purified protein is incubated. Upon washing, the presence (or absence) of the protein is revealed through a procedure reminiscent of a Western blot [178]. A similar possibility is the crosslinking of the lipid of interest to agarose beads, which are also incubated with the protein. These methods are fast and allow for scoring of multiple types of lipids on a single experiment, yet their major drawback is that the spotted lipids are not presented in their native state nor embedded in a membrane context.

To overcome this limitation, the alternative is to evaluate the binding of the purified protein to liposomes, engineered spheres conformed by a lipid bilayer harboring inside a small aqueous lumen. The bilayer can be designed to possess the proportion of carrier and problem lipids at will and comparisons made between those liposomes and the control ones, in which the problem lipid is omitted [179]. Furthermore, liposomes can also be created to possess a given size, thus a given diameter, which yields different surface curvatures. This way, smaller liposomes will present higher positive curvature, a master feature conditioning lipid-protein interactions [11]. Thus, depending on whether we suspect the lipid of interest as being either part of the core of condensate structures, or inserted in the INM, for example, the former or the latter technique may be more appropriate, respectively.

In both cases, though, the approaches allow assessing the binding of proteins capable of interacting with the polar head of the target lipids. A different situation is that of proteins with a deep hydrophobic pocket capable of hosting the apolar lipid tails [180], which the liposome assay may therefore render inaccessible. This binding in a pocket is reminiscent of and may be tackled as, the study of FABPs, which mostly requests heavy structural studies.

#### 3.2.2. Bioinformatic Prediction

Lipid-binding motifs define the parts of a protein with a specific affinity for a given lipid species. These include, among others, C1 (binds Diacylglycerols), PH (binds different types of Phosphoinositides), FYVE and PX (bind PI(3)P), C2 (binds different lipids and Phosphatidylserine in a Ca^2+^-dependent manner), Q2 (binds Phosphatidic acid), and PHD (binds PI(5)P) [181]. A comprehensive search for lipid-binding motifs in the whole subset of nuclear proteins is lacking. A bioinformatics search and experimental validation have already been done in *S. cerevisiae* for PX domains, where the analysis yielded no nuclear proteins [182]. For PH domains, 33 proteins were predicted, of which five have nuclear functions, and four are responsive to problems during DNA replication [183]. Initial predictions for many of these domains can be undertaken using SMART [184] or HMMER [185] tools. 

Another prediction tool concerns Fatty Acid-binding pockets, for which the size and shape of the entity render the enterprise less straightforward. An in-house built algorithm predicted the tumor suppressor BRCA1 as a candidate [76,186]. Unfortunately, this tool is no longer available. We, therefore, devised a trained detection algorithm using the sequence of a set of known FA-binding proteins (lipocalin family, up to now approximately 80 members of known 3D structure) and creating a PSSM (Position-specific score matrix). We next expanded it by aligning the initial 80 members with their counterparts from other species. This tool, called PyFuncover, was able to predict known FA-binding motifs that had not been used for the training [187]. A limitation was still the 3D aspect, given that FA-binding pockets can represent complex cavities. We have therefore created PickPocket, which uses neural networks to train a ligand-binding prediction model. Using FA-like ligands, we can define pocket descriptors and secondary structures with an accuracy above 90% by using a dataset of 1740 manually curated ligand-binding pockets. Pickpocket successfully predicts ligand-binding pockets using unseen structural data [188]. Using this tool, we have defined two central channel nucleoporins, Nup157 and Nic96, known to root the nuclear pore complex, as binders of FA, a feature that could regulate their dynamics [77].

Amphipathic helices (AHs) are a feature of the secondary structure of proteins in which the aminoacids fold in a helical manner that segregates hydrophobic and polar residues between the two faces of the helix. This spatial splitting permits inserting the concerned protein segment between the hydrophobic acyl chains and the polar heads of the phospholipids in a layer [189]. This apparently simplistic mechanism permits protein anchor to different organelles in a tuneable, reversible manner. For example, the interaction may be stabilized under conditions that permit the AH folding, while protein solubilization will occur if the surrounding medium is not favorable to the AH conformation. The properties of AHs vary depending on the size of hydrophobic residues as well as their density per helical turn, the type and distribution of the polar ones, and the overall length of the AH [190]. Reciprocally, the lipid composition of the surface to which the AH is likely to bind, or of the underlying lipid core in case of phospholipid monolayers, will dictate at a given moment the type and number of interacting AHs, as well as the tightness of their binding [191]. Bioinformatic prediction of AH can be tackled through tools such as MPEX (https://blanco.biomol.uci.edu/mpex/, accessed on 29 November 2021) or Amphipaseek [192], which are powerful to localize the relevant segments within candidate proteins suspected by the user. A complementary approach is provided by HELIQUEST, which allows the user to define the features of known helices and exploit them as a starting point to extract putative equivalent helices in unexpected proteins. Further, HELIQUEST helps devise inactivating mutations expected to disrupt the AH, allowing to assess whether this abolishes the protein interaction with the membrane [193].

Another relevant way of changing protein sub-compartmentalization is through transmembrane domains. This, for example, keeps transcription factors away from the promoters they could activate. Upon proteolytic processing of the protein, the transmembrane segment stays anchored into the membrane while the released peptide diffuses within the nucleus to accomplish its task [194,195]. Initial predictions of potential transmembrane helices in nuclear proteins can be tackled using Phobius and TMpred software. More reliable predictions will be achieved if a given fragment is highlighted by both tools.

Validation of bioinformatic predictions of putative lipid-related protein domains can be undertaken by cloning the motifs fused to a fluorophore of choice (see below). In the case of lipid-binding motifs, further confirmation can be obtained by cloning one-half of the Venus reporter protein fused to the candidate motif, while a nucleus-targeted lipid biosensor known to recognize the same lipid (see later) can be cloned fused to the other half of Venus. By virtue of BiFC (Bimolecular Fluorescence Complementation) [196], the Venus signal will be reconstituted only if the two elements of the system indeed stand close by within the nucleus, which will further inform of the sub-nuclear localization of the interaction. In this context, even if the biosensor competes with the protein fragment under test for lipid binding, the proximity of both within functional rafts, where many molecules gather, will make possible the detection of positive interactions. The feasibility of this strategy has been illustrated at the INM [130,197].

#### 3.2.3. Photoactivatable Lipids

A powerful and unbiased strategy to unveil non-covalent lipid/protein interactions relies on photoactivatable lipid analogs. These are synthesized lipids bearing a group that, upon short UV irradiation, will make a fast and irreversible reaction with adjacent proteins. The photoactivatable chromophore may be positioned either at the polar head or within the hydrophobic part of the molecule. The ideal photoactivatable group should be small as not to interfere with the physicochemical properties of the native biomolecule in which it is introduced, thus safeguarding the behavior of the intended lipid. It should also be stable under non-activating conditions, yet it should undergo rapid photolysis upon exposure to UV light, which limits long exposure times that may damage the biological sample. Further, these activated species should react in a fast and non-selective manner with molecules in their direct vicinity, limiting this way the false detections due to diffusion processes [198]. Apart from being photoactivatable, these lipids can simultaneously be “clickable”, named bifunctional lipids. Azide-biotin or azide-fluorophore click chemistry thus allows to subsequently study the UV-stabilized lipid-protein interactions in a manner as that explained for *bonafide* lipidation (Figure 2, right panel). In this context, an important control relies on comparing the results with those of equally processed samples in which UV irradiation is omitted [39].

### 3.3. Immunoprecipitation of Membranes 

Two main issues interfere with the goal of robustly defining the lipid composition of the nuclear membrane: first, the pure isolation of the nuclear membrane is virtually impossible, for it is generally accompanied by traces of the appended (continuous) cytosolic endoplasmic reticulum; second, no technique reliably permits discerning the outer from the INM. While not getting fully rid of these problems, an immune-enrichment approach can attempt to minimize them. The first step exploits a protocol for preparing high-purity nuclear envelopes [199]. In this protocol, rough ER components are virtually absent except for some residual ribosomes. Stretches of the nuclear membrane as long as 3 µm can be recovered, and electron microscopy has unequivocally validated the identity of outer and inner membranes in *S. cerevisiae* based on the positioning of the spindle pole body, anchored at the INM [199]. Moreover, the natural curvature of the nuclear membrane is respected (concave on the inner side, convex on the outer one), which speaks of the integrity of the composing lipids. The second step takes benefit from a given protein anchored to the membranes in order to enrich them by immunoprecipitation. The strategy for immuno-isolating membranes requests that the immunoprecipitated protein displays a strong link with the membrane, such as a lipidation moiety. In this case, an appropriate control relies on using cells deficient for that lipidation [200]. An INM transmembrane protein could also be exploited to isolate nuclear membranes, in general, and isolated (and refolded) INM segments, in particular. The protein Asi1 localizes to the INM with its C-terminus falling within the nucleus, and the GFP (green fluorescent protein)-tagging of this C-terminus does not alter the biology of the protein [197]. Thus, the use of the GFP-Trap^R^ technology (Chromotek), which exploits small recombinant alpaca antibody fragments covalently coupled to beads, would permit Asi1-GFP to be purified with high efficiency. The membranes obtained this way are suitable to be subjected to lipidomic analyses.

### 3.4. Seeing Lipids

Detecting lipids at their natural locations is challenging. On what concerns fixed samples, the use of antibodies is a poor option because of the difficulty in raising them in a specific manner, because the fixation that is needed prior to the use of antibodies may affect the lipid to be detected and because antibodies are significantly larger than most lipids, thus recognizing several lipid molecules at a time [201]. In this sense, only antibodies against PI(4,5)_2_P and PI(4)P have demonstrated reliability [98,109]. An alternative relies on biosensors. Biosensors are domains from proteins whose ability for binding a given lipid is well-characterized. These fragments are usually cloned adjacent to a fluorophore, thus permitting the visualization of the bound lipid [202,203,204]. It is possible to purify them for their exploitation antibody-wise through hybridization of fixed samples [205,206]. Alternatively, cloning them in appropriate vectors will permit the cell to express them. In this case, the signals can be visualized either upon fixation or during live microscopy. A drawback of this approach is the heterogeneity in the levels of expression among different cells. Moreover, their mid- and long-term expression are usually toxic, as the lipids that the biosensor binds become unavailable for their natural targets in the cell. As such, making stable cell lines, when not impossible, is not advised [97], and inducible systems should be envisioned. Nevertheless, transfection with biosensors permits short-term studies in which occlusion of the target lipid can be exploited for competition assays in comparison with an empty vector [207]. In all cases, the relevance of encouraging results obtained using a biosensor should ideally be compared with a similar construct in which specific aminoacids, known as responsible for the interaction, have been mutated [97]. Another note of caution relates to the fact that domains coming from different proteins and capable of recognizing one very same lipid can differ in the conditions under which the lipid is detected [208]. It is thus wise to use different known domains to probe for the same lipid. The yielded results, if common, will provide a robust readout of the actual lipid presence; if different, will orient the research towards other adjacent lipids presumably influencing the result. For example, phosphatidic acid detection is influenced by the actual presence of sterols in the membrane, thus being better detected by an Opi1-derived biosensor than by a Spo20-derived one when surrounding sterol levels are low [208].

While the use of biosensors is a well-established methodology, it rarely provides signals coming from the nucleus. Recent work has demonstrated that this is because the frequently higher pools of these lipids in other cellular membranes sequester the biosensor. Importantly, as a proof-of-concept for two particular biosensors, the same authors overcame this limitation by fusing the biosensor to a nuclear localization signal (NLS) (Figure 3A) [130]. The fusion of an NLS to the biosensors in a systematic manner should permit the creation of a collection of vectors, each allowing the expression of a fluorescent, nucleus-driven lipid biosensor for each relevant lipid species (sterols, Diacylglycerol, Phosphatidylinositol, PIP, Phosphatidic Acid and Phosphatidylserine). The different protein domains necessary to recognize and bind specifically these diverse lipids have been characterized before [202,203,204,205,209] (for a compilation, see [203]). This could yield a comprehensive collection of plasmids allowing us to ask ambitious questions: is a given lipid species enriched in the nucleus? Where exactly: floating in the nucleoplasm? (Figure 3B) At the INM? (Figure 3C). Does it localize to nucleoli or at bulk DNA? does it coincide with heterochromatin? Does its distribution change when cells are exposed to DNA-damaging agents? Or transcription inhibitors? Or how does it evolve throughout the cell cycle? Is the distribution the same in a mutant context of interest? Is it perturbed with the diet?

Last, another means of seeing lipids at their natural locations relies on the use of vital dyes or fluorescent analogs [201,210]. While in some cases, these may be water-soluble, simplifying the procedure, sometimes the delivery of these molecules to the cell may require some optimization [211]. In most cases, vital dyes can be used both with alive and fixed cells. A wide palette may be available for a similar purpose with respect to the excitation/emission choices, thus permitting a flexible combination with other visualization needs. For example, to visualize LD, AUTODOT^TM^ emits in blue, BODIPY^TM^ emits in green, and LipidTOX^TM^ emits in far red. It is important to remember, anyhow, that their long-term presence in the cell can alter the cell’s normal physiology. For fluorescent analogs, it is worth ensuring that the modification does not alter the lipid’s natural function and bio-physicochemical properties. It should also be remembered that the labeling represents a supplementation (thus a change) with respect to the basal amount of that lipid species available for the cell.

### 3.5. Purification of LD 

Purification of LD is a well-established method that takes advantage of the floatability of LD. In this way, upon depositing the extract at the bottom of a discontinuous sucrose gradient followed by centrifugation, LD will be found in the upper, most accessible phase [212]. The quality of the preparation can be assessed by direct inspection of the sample upon addition of a vital dye, such as Nile Red, thus providing an instantaneous picture of the integrity and (expected) size of the purified LD [122,138]. Additionally, a Western blot of the gradient fractions can be performed, where markers of LD *versus* those of other potentially contaminating fractions (especially the endoplasmic reticulum and plasma membranes) permit the assessment of the isolation quality [140,213]. By subsequently employing Mass Spectrometry and Lipidomics analyses, one can assess the unexplored proteome and lipidome, respectively, in the situation of interest. A vast repertoire of works has already assessed this in multiple situations and organisms, with a strong focus on metabolic conditions and on cytoplasmic pathways (for a compilation, see [121]). In comparison, only one work has explored the proteome of nuclear LD. Yet, in their attempt for stringency and given the difficulty in obtaining enough starting material, the presented list of currently found proteins is short [143]. Overall, little is known about the regulatory activity of cytoplasmic and nuclear LD as sinks for factors relevant in nuclear homeostasis and, even less, under different nucleus-challenging conditions.

## 4. Concluding Remarks

The intertwining of genome integrity and metabolism (to which lipids are central) when it comes to pathology arousal, treatment and relapse is nowadays recognized. Thus, developing our understanding of the links between the metabolism of lipids and genome integrity bears a strong potential to gain insights into the pathophysiology of genetic, rare, and degenerative diseases, including envelopathies, lipodystrophies, atherosclerosis, hepatic steatosis, and neurodegenerative conditions, as well as cancer, viral infections, or even healthy *versus* pathological aging. Given that the research about the impact of lipids in genome stability maintenance is at its infancy, a collective effort will be essential to push our knowledge further.

## Figures and Tables

**Figure 1 ijms-22-12930-f001:**
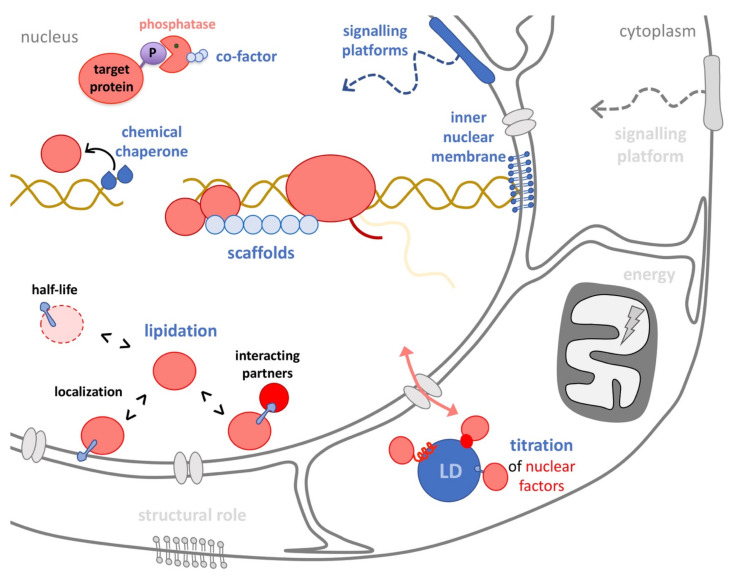
Overview of how lipids could impact nuclear transactions. Classical, well-studied functions of lipids in the cytoplasm, such as membranes constituents, as energy suppliers, or as plasma membrane-based signaling platforms, are depicted in grey. Less studied and even hypothetical activities in the nucleus have been drawn in blue, with the immediately concerned nucleic acids in yellow and proteins in red. These activities include structuring and signaling roles at the inner nuclear membrane, scaffolding of complexes, activating roles as cofactors, dislodgement of proteins by chemical chaperoning, lipidation of nuclear proteins driving changes in their interactome, subnuclear localization or half-life, as well as indirect titration of nuclear factors in the cytoplasm by their anchoring to the surface of lipid droplets (LD).

**Figure 2 ijms-22-12930-f002:**
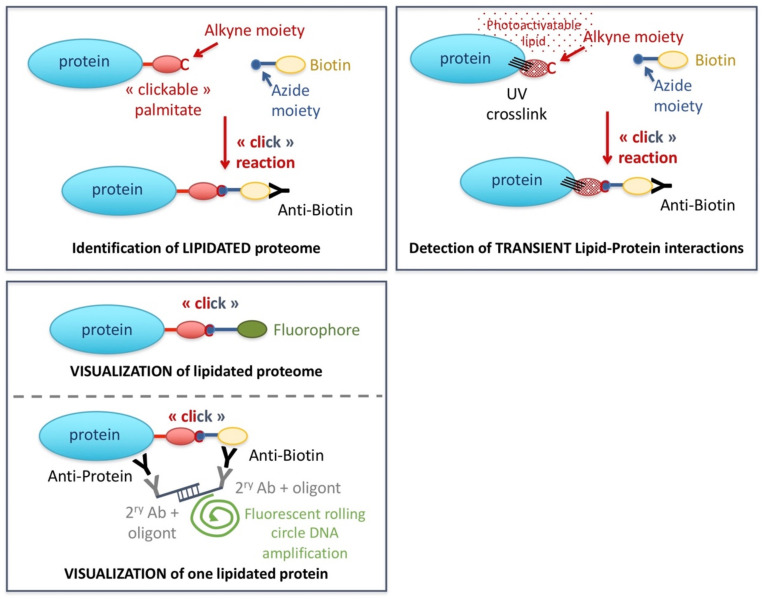
The use of commercially available clickable lipids allows the isolation of proteins covalently modified by lipids (**left**), interacting with lipids in a transient fashion (**right**), and also to follow the subcellular distribution of those lipids visually by fluorescence microscopy (**bottom left**).

**Figure 3 ijms-22-12930-f003:**
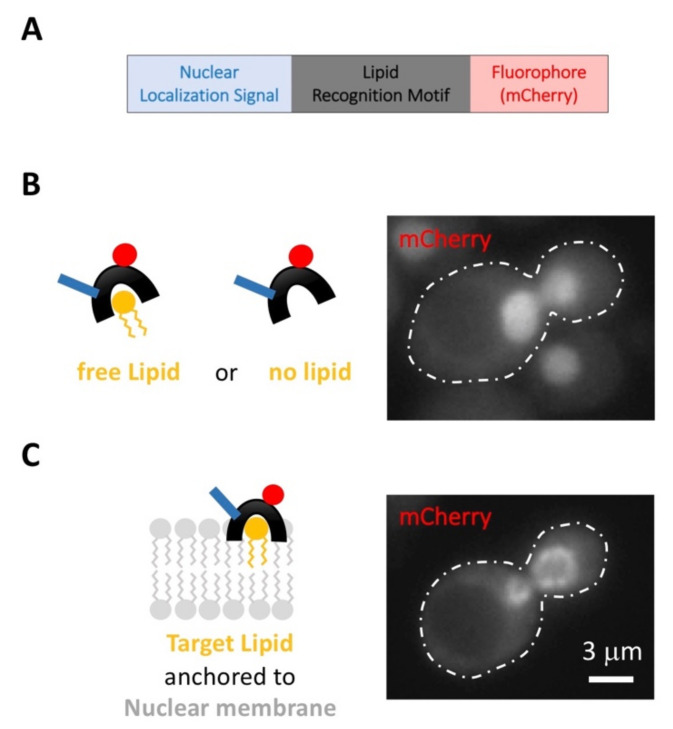
The use of nucleus targeted-fluorescent lipid biosensors permits the monitoring of how specific lipid species evolve under a myriad of experimental situations. (**A**) Basic structure of a nucleus-targeted biosensor. (**B**) Hypothetical example in which the mCherry signal associated with the biosensor emanates from the nucleoplasm of *S. cerevisiae* cells, which can be ascribed either to a floating lipid or to the absence of any binding target. (**C**) Hypothetical example in which the mCherry signal associated to the biosensor emanates from the nuclear membrane, indicative of either an enrichment of the lipid of interest at this location or, at least, of its exposure.

## Data Availability

Not applicable.

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
