# Peer review of "The Many Faces of Lipids in Genome Stability (and How to Unmask Them)"

_ijms, 2021, doi:10.3390/ijms222312930_

Round 1

Reviewer 1 Report

The author reviewed the literature about the possible role of lipids in the stability of genome integrity. The topic is interesting because of the relatively small number of relevant studies.

Some abbreviations are explained in the text, but many of them are not introduced. I don’t understand what determined that something was explained, and others not. For example, FABP (row 181) and CTBP (row 187) are explained, but PPAR and FAAR are not (row 184). The text is full of abbreviations, but only 7 of them are listed at the end in ‘Abbreviations’ section. Please write down all abbreviations to help readers, who are not so familiar with lipids, genome and these abbreviations.

On page 5, ‘fatty acids’ and ‘ceramides’ are bold, but I don’t understand, why. When you want to highlight the lipid currently discussed in the text, why are the next lipids not bold (e.g. phosphoinositides)?

From the text (row 167) it seems to me that acetyl-group (C2) is the common, collective name for all short-chain fatty acids (C1-C6), although it only means the fatty acid with two carbon atoms. Please correct it.

There are many data about the possible roles of different lipids in different cells (bacteria, animals and human). Maybe a Table would give a better overview, which of these pathways are only found in single-celled organisms, which are relevant for animal cells also and how many of them were proved in human cells also.

There are many errors and missing parts among the references. I list many of them here, but please double check all the references:

  1. Nr.26: Kidiyoor et al, ‘ATR is essential for preservation of cell mechanics’ is missing from the beginning of the title. The article number and publication year is also missing (Nat Commun 2020, 11, 4828). Please correct.
  2. In several references, Volume and page numbers are lacking (9, 14, 19, 28, 30-35, 40-42, 52, 87, 95, 97, 103, 106, 115-122, 124-127, 130, 133-138, 141, 154, 173,183, 186, 192, 194, 196, 199, 201-203, 207-208)
  3. In some references DOI numbers are also lacking (30-32, 95,115, 127, 139-140, 169)
  4. Nr.64: Title is wrong, Volume, page numbers, DOI are lacking (Liau G, Ong DE, Chytil F. Interaction of the retinol/cellular retinol-binding protein complex with isolated nuclei and nuclear components. J Cell Biol. 1981;91(1):63-8. doi: 10.1083/jcb.91.1.63.) Please correct.
  5. Nr. 189: Title doesn’t seem good, and I also couldn’t find this doi, please check it.
  6. Nr. 191: Title is missing, doi is incorrect, Kerppola TM is the only author, please correct it.
  7. Nr. 202: letter ‘a’ is missing from the first author’s name (instead of Várni P, Várnai P)

Author Response

Reviewer 1 :

 The author reviewed the literature about the possible role of lipids in the stability of genome integrity. The topic is interesting because of the relatively small number of relevant studies.

I am very thankful to this reviewer for the very thorough reading and analysis of my manuscript, as evidenced not only by the pertinent suggestions but also by the depth of the raised details. I answer to him/her after each concern, identifying my reply by making it be preceded by the symbol “>”:

Some abbreviations are explained in the text, but many of them are not introduced. I don’t understand what determined that something was explained, and others not. For example, FABP (row 181) and CTBP (row 187) are explained, but PPAR and FAAR are not (row 184). The text is full of abbreviations, but only 7 of them are listed at the end in ‘Abbreviations’ section. Please write down all abbreviations to help readers, who are not so familiar with lipids, genome and these abbreviations.

  • The reason to include some abbreviations in the “abbreviations” sections and some others not relies on the frequency of use. For example, “PIP” (phosphoinositides) is used recurrently, thus it was included in the abbreviations section; but PPAR, for example, was used only once, thus it was not. I have kept this reasoning in the revised version but I have tried to write the full name side by side with its acronym every time. Thank you, it is clearer now.

On page 5, ‘fatty acids’ and ‘ceramides’ are bold, but I don’t understand, why. When you want to highlight the lipid currently discussed in the text, why are the next lipids not bold (e.g. phosphoinositides)?

  • The reviewer is right. There was an inconsistency here. Now there is no bold for any lipid species.

From the text (row 167) it seems to me that acetyl-group (C2) is the common, collective name for all short-chain fatty acids (C1-C6), although it only means the fatty acid with two carbon atoms. Please correct it.

  • I fully agree. The way the notion was expressed has been changed accordingly, I think it is not misleading any more.

There are many data about the possible roles of different lipids in different cells (bacteria, animals and human). Maybe a Table would give a better overview, which of these pathways are only found in single-celled organisms, which are relevant for animal cells also and how many of them were proved in human cells also.

  • This was a fantastic suggestion. I have included a table now (Table 1, included in page 11), in which I have recapitulated the up-to-date known role of lipids in nuclear transactions. I have classified in the first column the “action” exerted in the nucleus, respecting the view I provide throughout the full section 2. Then, each “action” splits into the different type of lipid species, the organism(s) where the findings were made and the associated reference(s). Thank you for this valuable suggestion.

There are many errors and missing parts among the references. I list many of them here, but please double check all the references:

  • Overall, this was a big issue derived from annotation mistakes in my formatting tool. I have triple-checked, and I think it is now corrected. The only exception I am aware of concerns reference 107, for which I could not find any DOI. I hope the overall problem is fixed now. Thank you for being so meticulous.
  1. Nr.26: Kidiyoor et al, ‘ATR is essential for preservation of cell mechanics’ is missing from the beginning of the title. The article number and publication year is also missing (Nat Commun 2020, 11, 4828). Please correct.
  2. In several references, Volume and page numbers are lacking (9, 14, 19, 28, 30-35, 40-42, 52, 87, 95, 97, 103, 106, 115-122, 124-127, 130, 133-138, 141, 154, 173,183, 186, 192, 194, 196, 199, 201-203, 207-208)
  3. In some references DOI numbers are also lacking (30-32, 95,115, 127, 139-140, 169)
  4. Nr.64: Title is wrong, Volume, page numbers, DOI are lacking (Liau G, Ong DE, Chytil F. Interaction of the retinol/cellular retinol-binding protein complex with isolated nuclei and nuclear components. J Cell Biol. 1981;91(1):63-8. doi: 10.1083/jcb.91.1.63.) Please correct.
  5. Nr. 189: Title doesn’t seem good, and I also couldn’t find this doi, please check it.
  6. Nr. 191: Title is missing, doi is incorrect, Kerppola TM is the only author, please correct it.
  7. Nr. 202: letter ‘a’ is missing from the first author’s name (instead of Várni P, Várnai P)
  • I want to additionally let the reviewer know that there are changes made to the text, and they can be recognized because they are indicated in green font (except the table), in order to facilitate the task of identifying them. The reasons why some changes were done relate to the inclusion of the meaning of the abbreviations; to splitting of some sentences, as requested by Reviewer 3; or to including new pertinent bibliography that was not present in the previous version either because of my inadvertence, or because the concerned works were just recently published.

Reviewer 2 Report

Dear Editor,

I carefully read the manuscript by Moriel-Carretero. Unfortunately, I have serious concerns as regards the quality of the current article from a scientific point of view.

Moriel-Carretero does not explain in the manuscript how the articles included among the references were selected. As a results, this paper looks more like a "Point of view" of the author than a literature review.

Figures are very accurate.

Author Response

Reviewer 2:

I carefully read the manuscript by Moriel-Carretero. Unfortunately, I have serious concerns as regards the quality of the current article from a scientific point of view.

Moriel-Carretero does not explain in the manuscript how the articles included among the references were selected. As a results, this paper looks more like a "Point of view" of the author than a literature review.

  • The manuscript aimed at gathering the available scientific information, if any, that could support the notion that lipids are important for nuclear homeostasis, in general, and genome integrity, in particular. As such, the criteria for the selection of the included references was exclusively related to fulfilling this mission. I was not aware that there was a specific need for justifying this further when using such references. Else, the impression that the text may look like a “point of view” was intended to be restricted to some precise locations of the text in which, overtly, I indicated that that was speculation, the goal simply being to raise potentially interesting questions that may seed further research. I tried therefore to make a clear distinction between what is actually known, and what is just a yet-to-demonstrate possibility. Last, I am particularly concerned about the reviewer’s opinion that the article does not hold “quality from a scientific point of view”. The document was written under a scrupulous respect to the work of others, the description of their findings in an objective manner and the clarification of what is known from what it is not whenever I dared speculation, as just explained. Overall, other than providing the reviewer with these arguments, I have difficulties to assess, thus to improve, my document in view of the current criticism. I sincerely hope that the changes I implemented after the suggestions from Reviewers 1 and 3 serve to ameliorate the perception of this Reviewer 2.

Figures are very accurate.

  • Thank you very much for this positive appreciation.

Reviewer 3 Report

This manuscript comprises an excellent review of yet poorly understood significance of Lipids in genome maintenance. Dr. Moriel-Carretero has compiled the relevant literature in the field in a very systematic way, highlighting significant discoveries, existing gaps and posing new questions. The manuscript also describes various techniques and experimental strategies to investigate lipids in the context of genome homeostasis. Manuscript text is ably complemented with 3 very nice figures. Overall, I really enjoyed reading this manuscript, and believe this review to be an excellent resource to the wider scientific community. 

Minor suggestion: I think the language/English can be revised to improve readability. For example, the first sentence in the abstract can be written in a simpler style -I had to look up the meaning of "consecrated" and don't understand the intended meaning. There is a tendency to use too many commas (,) throughout the manuscript, and that is somewhat distracting. Perhaps long sentences can be broken down to simpler sentences.

Round 2

Reviewer 2 Report

Dear Editor,

I am very sorry to have to confirm a negative opinion on Dr. Moriel-Carretero article, but unfortunately I really do not think it deserves to be published. Briefly, I do not think that this article is neither useful nor of any interest for the readers of the Journal. I recognize that the figures are very accurate, however it continues to be unclear how the references were selected so that the manuscript looks more like a long editorial than a review paper. Dr. Moriel-Carretero is does not have experience enough to write a  point-of-view article (SCOPUS h-index= 6) so that I cannot suggest to publish this manuscript in the Journal.